# The Impact of Acetyl-CoA and Aspartate Shortages on the *N*-Acetylaspartate Level in Different Models of Cholinergic Neurons

**DOI:** 10.3390/antiox9060522

**Published:** 2020-06-13

**Authors:** Marlena Zyśk, Monika Sakowicz-Burkiewicz, Piotr Pikul, Robert Kowalski, Anna Michno, Tadeusz Pawełczyk

**Affiliations:** 1Department of Molecular Medicine, Medical University of Gdansk, 80-211 Gdansk, Poland; monika.sakowicz-burkiewicz@gumed.edu.pl (M.S.-B.); tadeusz.pawelczyk@gumed.edu.pl (T.P.); 2Laboratory of Molecular and Cellular Nephrology, Mossakowski Medical Research Center, Polish Academy of Science, 80-308 Gdansk, Poland; piotr.pikul@gumed.edu.pl; 3Clinical Laboratory University Clinical Center in Gdansk, 80-211 Gdansk, Poland; robert.kowalski@gumed.edu.pl; 4Department of Laboratory Medicine, Medical University of Gdansk, 80-2011 Gdansk, Poland; anna.michno@gumed.edu.pl

**Keywords:** 2-APB, zinc neurotoxicity, diabetes

## Abstract

*N*-acetylaspartate is produced by neuronal aspartate N-acetyltransferase (NAT8L) from acetyl-CoA and aspartate. In cholinergic neurons, acetyl-CoA is also utilized in the mitochondrial tricarboxylic acid cycle and in acetylcholine production pathways. While aspartate has to be shared with the malate–aspartate shuttle, another mitochondrial machinery together with the tricarboxylic acid cycle supports the electron transport chain turnover. The main goal of this study was to establish the impact of toxic conditions on *N*-acetylaspartate production. SN56 cholinergic cells were exposed to either Zn^2+^ overload or Ca^2+^ homeostasis dysregulation and male adult Wistar rats’ brains were studied after 2 weeks of challenge with streptozotocin-induced hyperglycemia or daily theophylline treatment. Our results allow us to hypothesize that the cholinergic neurons from brain septum prioritized the acetylcholine over *N*-acetylaspartate production. This report provides the first direct evidence for Zn^2+^-dependent suppression of *N*-acetylaspartate synthesis leading to mitochondrial acetyl-CoA and aspartate shortages. Furthermore, Zn^2+^ is a direct concentration-dependent inhibitor of NAT8L activity, while Zn^2+^-triggered oxidative stress is unlikely to be significant in such suppression.

## 1. Introduction

Aspartate N-acetyltransferase (NAT8L) is a neuronal enzyme producing *N*-acetylaspartate (NAA) from acetyl-CoA and aspartate (Figure 1A) [1,2,3]. Therefore, in cholinergic neurons, NAA synthesis has to share acetyl-CoA with the tricarboxylic acid cycle and the acetylcholine production pathway, while aspartate is shared with the aspartate—malate shuttle [2,3,4,5]. Both the tricarboxylic acid cycle and the malate–aspartate shuttle are the main participants of the mitochondrial machinery supporting electron transport chain turnover. Acetylcholine synthesis is exclusively reported in the cholinergic neurons as a cytosolic setup reaction starting the cholinergic neurotransmission chain (Figure 1A) [6,7]. *N*-acetylaspartate is an exceptional messenger in the crosstalk takes place between neurons and glial cells, such as oligodendrocytes or astrocytes [5,8]. Other words, NAA as well as acetylcholine are produced to fulfill the neuronal functions linked with cell-cell interactions. Moreover, they have to share substrates with two powerful energy machineries supporting regular cell functions [6]. Thus, this unusual situation poses the crucial question: which of these metabolites (NAA, acetylcholine) will have priority in situation of acetyl-CoA shortages?

Recently, aspartate N-acetyltransferase (NAT8L) has been identified as a candidate for a mechanism-based molecular target in treatment strategies against neurodegeneration concomitant with progressive brain disorders starting with degeneration of cholinergic neurons [9]. The newest research framework for such neurodegeneration treatment pointed the neuronal energy disorders as the most urgent goals [9]. A working hypothesis claims that the inhibition of NAT8L activity will reduce acetyl-CoA consumption in the *N*-acetylaspartate production pathway. Consequently, released acetyl-CoA may support neuronal energy production. However, knowledge about the factors regulating NAT8L activity is rather poor and there is no guarantee that such inhibition will enhance energy production or will not be lethal. Considering the unusual acetyl-CoA metabolism noted in cholinergic neurons (extra utilization pathway producing the acetylcholine neurotransmitter, Figure 1A), we wonder if NAT8L inhibition will affect homeostasis in the acetyl-CoA utilization rate which takes place in energy production, acetylcholine production as well as NAA production.

Our previous studies showed that the chronic exposure of SN56 cells (cellular model of cholinergic neurons) to Zn^2+^ overload may cause a concentration-dependent NAA-reduction as well as partial NAT8L inhibition [4]. However, the exact mechanism regulating the NAA production rate has not yet been established. Here, we considered substrate shortages, high cholinergic phenotype expression, direct Zn^2+^ inhibition of NAT8L or indirect impact (Zn^2+^-triggered free radical overload) [4,10,11,12]. Therefore, the main goal of this study was to establish, which of these possibilities plays the main role in Zn^2+^-dependent NAA reduction as well as to investigate the relationships between the malate–aspartate shuttle, the tricarboxylic acid cycle, cholinergic neurotransmission and *N*-acetylaspartate production.

To study the relationship between NAA and acetylcholine levels, we used either an in vitro model of cholinergic neurons (SN56 cells) or investigated the particular rats’ brain regions. The brain septum was chosen as the area with the densest occurrence of cholinergic neurons and the brain region from which the SN56 cell line was isolated [13], while the cerebellum was used as a negative control having significantly lower choline acetyltransferase activity. Our previous study focused only on the relations between NAA level and acetyl-CoA shortages [4]. This time, we considered the impact of both acetyl-CoA/aspartate shortages as well as cholinergic phenotype presentation on the NAA level and NAT8L activity. The SN56 cells were treated with 0.15 mM Zn^2+^ to induce either chronic or acute toxicity. The acute approach showed energy disorders evoked by Zn^2+^ uptake taking place under depolarizing conditions [12]. To distinguish different Zn^2+^-related biochemical changes, we confirmed them using different inhibitors: mecamylamine to suppress cholinergic neurotransmission [14], nifedipine to control calcium influx and free radical production [12] as well as 2-aminoethoxydiphenyl borate to affect ATP level [15]. The chronic Zn^2+^ treatment showed the long-term impact on the NAT8L activity [4]. Studies with Zn^2+^ treatment showed a significant impairment in acetylcholine release reflecting disorders in cholinergic neurotransmission. To investigate whether in brain the affected cholinergic neurotransmission accompanies NAT8L inhibition, we challenged the animals by streptozotocin or theophylline treatment affecting the cholinergic neurons. Streptozotocin-induced neurodegeneration differs from theophylline—triggered cholinergic disorders. The hyperglycemic animals presented free radical overproduction, which strongly imitates Zn^2+^-induced toxicity, while theophylline forces cyclic AMP accumulation showing side effects of protein kinase A activation.

## 2. Materials and Methods

### 2.1. Materials

Unless specified otherwise, all the used compounds were specified at Appendix A, while cell culture disposables were provided by Sarstedt (Blizne Łaszczyńskiego Poland). Unless specified otherwise, spectrophotometric assays were run either using Ultraspec 3100 Pro (Amersham Biosciences, Amersham, UK) or, for multiple well plate-based assays, Victor 3, 1420 Multilabel Counter (Perkin Elmer, Warsaw, Poland).

### 2.2. Animals

All experiments were approved by the Polish Bioethics Committee (44/2016, 23 November 2016, Bydgoszcz, Poland). Studies followed the EU Directive 2010/63/EU and the International Council for Laboratory Animal Science (ICLAS) guidelines for animal experiments.

Male adult Wistar rats (RRID: RGD_13508588) were housed at the Animal House (Medical University of Gdansk, Gdansk, Poland) with access to food and water ad libitum under a standard 12 h light/12 h dark cycle. The rats’ average weigh before the experiments was 180–230 g followed by 190–330 g of weight reached at the end of the experiments (Table 1). For this study purpose, animals were divided randomly to different treatment groups with the following group size: sham control group—11 male rats; diabetes mellitus group—8 male rats; theophylline group—8 male rats.

Diabetes mellitus (DM) was induced by a single intraperitoneal injection of 65 mg/kg BW streptozotocin (STZ) in 0.1 M citrate buffer (*ip*), while the theophylline treatment was provided by daily injections of 20 mg/kg theophylline/0.9% NaCl (*ip*). The sham control group was treated with a similar volume of pure buffer. After 2 weeks of studies, the animals were euthanized by the pentobarbitone overdose (2 mL/kg BW, ip, concentration: 0.66 M). All of used concentrations were used in accordance with either our previous projects or recommended by other researchers [17,18]. Brains were rapidly removed from euthanized animals and placed in ice-cold PBS buffer (pH = 7.4). Brain septum and cerebellum were dissected using the atlas of Swanson [16] as a guide for a tissue dissection (Figure 1B). To identify the brain septum area, we made the coronal section at the frontal brain (as shown at Figure 1B, the right upper corner), thus the striatum and septum became visible. The brain septum region was identified by its specific morphology and placed in buffers described below (Sample preparation).

Glycemia was measured at third and fourteenth day after streptozotocin administration in the animals’ blood serum (Accu-CHEK Performa glucometer kit, Roche, Warsaw, Poland). The animals with blood glucose ≥17 mM were deemed diabetic and suitable for this study [17].

### 2.3. Cell Culture

SN56 cells (murine neuroblastoma, RRID: CVCL_4456) were cultured for 48 h in DMEM supplemented with 2 mM L-glutamine, 0.05 mg/mL streptomycin, 50 U/mL penicillin and 10% fetal bovine serum. To set neuronal differentiation, culture media were enriched with mixture of 0.001-mM *trans*-retinoic acid with 1 mM dibutyryl cAMP. For the chronic studies, the SN56 cells were cultured for an additional 24 h in fresh media with or without 0.15 mM zinc chloride (ZnCl_2_, Zn^2+^).

For acute studies, 2–3 × 10^6^ cells were transferred to 1 mL of incubation media containing 2.5 mM pyruvate, 2.5 mM L-malate, 2.5 mM α-ketoglutarate and 2.5 mM glutamate, 20 mM sodium-HEPES (pH 7.4), 1 mM CaCl_2_, 1.5 mM sodium/potassium–phosphate buffer (pH 7.4), 32 mM sucrose and 30 mM KCl and 90 mM NaCl. To block particular calcium transporters by specific antagonists, media were supplemented by either 0.01 mM nifedipine (NF) or 0.050 mM 2-aminoethoxydiphenyl borate (2-APB) or 2 µM mecamylamine (MEC). While to establish highly neurotoxic conditions, the cells were treated by 0.15 mM ZnCl_2._ The experimental step was carried out for 30 min, 37 °C, with gentle shaking.

The concentration of 015 mM Zn^2+^ was identified in our previous projects as a highly neurotoxic factor in both chronic and acute experimental outlines [4,10,11,12].

### 2.4. Sample Preparation

Brain tissues (brain septum or cerebellum) or the SN56 cells were homogenized in chilled: 4% HClO_4_ (for metabolic studies); 0.1 M HCl (for NAD assay); 0.2 M KOH (for NADH assay); 5% metaphosphoric acid (for glutathione assays); buffer 5-mM HEPES (pH = 7.4) with 0.32 M sucrose and 0.1 mM EDTA (for enzymatic assays). After centrifugation at 13,000× *g* (4 °C, 15 min), each sample was immediately used for studies or kept at −80 °C until analyzed.

### 2.5. Mitochondria Isolation

SN56 cells were lysed for 30 s in mitochondrial isolation buffer (0.14 mg/mL digitonin, 125 mM KCl, 20 mM HEPES (pH = 7.4), 3 mM EDTA), then layered on AR20/AR200 oil mixture (1/2, v/v) and finally spun down (30 s, 14,000× *g*). The achieved pellet was identified as mitochondrial fraction [19].

### 2.6. Enzymatic Assays

To analyze enzymatic activity in the cell lines, from each dish two independent cell lysates were collected and reported as a one average result. To analyze enzymatic activity in brain tissue, 2 tissue samples were lysed per each brain region and then reported as a one average result. Protocols with details for enzymatic assays are shown in Appendix A.

Aconitase (Aco, EC 4.2.1.3) [20], aspartate aminotransferase (GOT, EC 2.6.1.1) [21], aspartate N-acetyltransferase (NAT8L, 2.3.1.17) [22], choline acetyltransferase (ChAT, EC 2.3.1.6) [23], citrate synthase (SC, EC 4.1.3.7) [24], glutamate dehydrogenase (GDH, EC 1.4.1.2) [25], Hexokinase (Hex, EC 2.7.1.1) [26], isocitrate dehydrogenase (IDH, EC 1.1.1.42) [27], lactate dehydrogenase (LDH, EC 1.1.1.27) [28], Pyruvate dehydrogenase complex (PDHC, EC 1.2.4.1.) [29], malate dehydrogenase (MDH, EC 1.1.1.37) [30].

### 2.7. Metabolic Assays

To analyze the level of particular metabolite in the cell lines, from each dish two independent acidic supernatants were collected and reported as a one average result. To analyze the metabolite level in brain tissue, 2 tissue samples were deproteinized per each brain region and then reported as a one average result. To analyze utilization rate, we collected 2 types of samples per each experimental point: the sample “zero”, which was deproteinized just before the incubation was started and the sample “stop”, which was deproteinized right after the incubation was completed. Next, result obtained from sample “stop” were subtracted from results obtained from respective sample “zero”. Protocols with details for enzymatic assays are shown in Appendix A.

Acetylacetate [31], acetyl-CoA [19], acetylcholine release [31], aspartate [32], ATP [33], β-hydroxybutyrate [31], lactate [31], malate [32], *N*-acetylaspartate [4], oxaloacetate and pyruvate [32], thiobarbituric acid reactive substances (TBARS) [34].

### 2.8. Western Blot Analysis

SN56 cells or its fractions were lysed for 30 min in lysis buffer (1% protease inhibitor cocktail, 50 mM Tris-HCl buffer (pH 7.4), 5 mM EDTA, 100 mM NaCl, 1% Triton-X100, 5% glycerol, 10 mM KH2PO4), at 4 °C. The obtained lysates were kept in −20 °C until analysis. Each sample (40 µg of protein/20 µL of 2.5% β-mercaptoethanol/Laemmli buffer) was boiled for 5 min and then loaded to Mini-PROTEAN 4–20% gradient SDS-PAGE TGX^TM^ gels (BioRad, Cat #4561093, Warsaw, Poland). Next, the SDS-PAGE gel was run at 300 V for 15 min in SDS running buffer (25 mM Tris (pH = 8.3), 190-mM glycine, 0.1% SDS). Next, proteins were transfer from to a nitrocellulose membrane (NC, pore size: 0.2 µm, iBlot^®^ transfer stack, Cat #IB301001, Warsaw, Poland) using iBlot^®^ Dry Blotting System with P0 program (program details: 1 min—20 V, 4 min—23 V, 3 min—25 V) (ThermoFisher Sc., Dreieich, Germany). The NC membrane was washed 2 × 10 min in TBTS buffer (25 mM Tris-HCl (pH = 7.4), 135-mM NaCl, 3-mM KCl, 0.5% Tween20). Non-specific bindings were blocked with 5% BSA in TBST (60 min, room temperature). Then, the NC membrane was incubated with specific primary antibodies (either rabbit anti-GADPH from Abcam, Cat #ab8245 or rabbit anti-KDHC from Santa Cruz Biotechnology, Cat #sc-67238, Heildelberg, Germany) in 5% BSA/TBST buffer (4 °C, overnight). At the following day, after 3 × 10 min washing with TBST buffer, the NC membrane was incubated with HRP-conjugated secondary anti-rabbit antibody (Sigma-Aldrich, Cat #A0545, Poznań, Poland) in 5% BSA/TBST buffer (3 h, room temperature). Finally, the NC membrane was developed by using the ClarityTM ECL western blot developing solutions (BioRad, Cat #1705060) and ChemiDoc System (Bio-Rad Laboratories, Warsaw, Poland) [12].

### 2.9. Real-Time RT-qPCR Analysis of NAT8L mRNA Levels

The 0.1 g of brain tissues were vortexed or homogenized in a sterile tube with 0.5 mL (cells) or 1 mL (brain tissue) of RNA extracol extraction buffer (Eurx, Cat #E3700-02). The extraction was initiated by the addition of 250 μL chloroform (per 1 mL of RNA extracol buffer). After vigorous shaking, each sample was incubated at 4 °C for 15 min and spun down (10,000× *g* for 15 min at 4 °C). The upper aqueous phase was transferred to a new tube and refilled by isopropanol (POCH, Cat #751500111) in ratio 1:2 (isopropanol: RNA extracol, v/v). The RNA precipitation was carried out overnight at −20 °C and on the following day each sample was centrifuged (10,000× *g* for 15 min at 4 °C). RNA pellet was washed first with 99.8% and then with 75% ethanol, finally air-dried and reconstituted in nuclease-free water (15–20 μL) (Sigma-Aldrich, Cat# W4502, Poznań, Poland). The obtained samples were kept at −20 °C until analyzed. The quantity of isolated RNA was determined using the Qubit RNA HA assay kit according to the manufacturer’s instructions (ThermoFisher Sc., Cat #Q32855, Warsaw, Poland). The gene expression levels of *NAT8L* encoding NAT8L enzyme was determined by real-time RT-qPCR performed in a Light Cycler 480 II (Roche Diagnostics GmbH, Penzberg, Germany) using Path-IDTM Multiplex One-Step RT-PCR Kit (ThermoFisher Sc., Cat #4442135, Warsaw, Poland) and Universal ProbeLibrary for the rat species and gene-specific intron-spanning primers (Table 2). The reaction mixture in the final volume 10 μL contained 5 μL of multiplex RT-PCR buffer, 1 μL of Multiplex Enzyme Mix and 0,5 μL of each primer for target transcript, 0,2 μL of a target probe, 0,2 μL of primers’ reference gene, 0,2 μL of probe for reference transcript and 2 μL of total RNA (Table 2). The target gene transcript levels were normalized to reference transcript of the β-actin gene (*Actb*). Reverse transcription program: 48 °C—10 min and 95 °C—10 min. Amplification program: 95 °C—15 s, 60 °C—45 s for 45 cycles. Data were processing with the Light Cycler 480 II software 2.0 [35].

### 2.10. Protein Assay

Protein was assayed by the method of Bradford with human immunoglobulin as a standard (standard curve: 0.2–0.8 mg/mL) [36].

### 2.11. Statistics

The results are presented as a median (25th–75th percentile). The Kolmogorov–Smirnov normality test exclude the normal data distribution. Therefore, the results were tested by either Mann–Whitney test or Kruskal–Wallis followed by Dunn’s multiple comparison post-test, where values of *p* < 0.05 were considered statistically significant. We performed all statistical analyses using the Graph Pad Prism 4.0 statistical package (Graph Pad Software, San Diego, CA, USA).

## 3. Results

### 3.1. Cholinergic Phenotype in SN56 Cell Line and Wistar rats’ Brain

Our previous studies showed a significant impact of Zn^2+^-related toxicity on the expression of cholinergic phenotype as well as on the level of *N*-acetylaspartate (NAA) in SN56 cells (cellular model of cholinergic neurons) [4,10,11,12]. The exact influence of Zn^2+^ on NAA production has not been established yet, although it is known that Zn^2+^ affects pyruvate dehydrogenase activity leading to acetyl-CoA shortages in SN56 cells [4,10,11,12]. Considering the biochemical background in which acetylcholine and NAA share acetyl-CoA as a substrate, we assumed that one of the Zn^2+^-dependent suppressive influence is linked with the acetyl-CoA shortages (Figure 1A). Other possibilities are linked with the aspartate N-acetyltransferase (NAT8L) activity, potentially by direct Zn^2+^ inhibition or by triggering excessive enzyme oxidation [4]. Therefore, in this study, we used 2 different approaches establishing the conditions affecting NAA production.

The in vitro approach assumed exposure of the SN56 cells to Zn^2+^ either chronically (24 h in media with 10% fetal bovine serum) or briefly (30 min, depolarizing serum-free media). To establish if acetyl-CoA shortages have the same impact on both NAA and acetylcholine productions, SN56 cells were exposed to mecamylamine (Mec, an antagonist of the nicotinic acetylcholine receptor), nifedipine (NF, an antagonist of the L-type voltage calcium channel) or 2-aminoethoxydiphenyl borate (2-APB, an antagonist of the IP3 receptor). Mec was used to reduce acetylcholine release [14], nifedipine to reduce free radical production [12] and 2-APB to evoke ATP shortages [15].

The in vivo approach studied the impact of either cholinergic neurotransmission or oxidative stress. To study the relationship between NAA level and cholinergic neurotransmission, the rats were challenged for 2 weeks by theophylline. Theophylline treatment forced prompt acetylcholine secretion and when it was used chronically it led to the impairment of cholinergic neurotransmission caused by excessive acetylcholine exhaustion [37]. Subsequently, streptozotocin-induced hyperglycemia was used as another model for presenting disorders in cholinergic neurotransmission, this time caused by the downregulation of choline acetyltransferase [38]. Furthermore, induced hyperglycemia is known as a suitable model concomitant with the upregulation of oxidative stress markers (Table 1) [39].

We noted that short-term exposure to mecamylamine did not change acetylcholine release in SN56 cells, while Zn^2+^ treatment reduced such release by about 50% (Figure 2A). Nifedipine and 2-APB significantly enhanced acetylcholine release by about 20% (Figure 2A). Since 30 min of mecamylamine treatment occurred to be insufficient to dysregulate choline neurotransmission, the animals were challenged for 2 weeks—either by streptozotocin-induced hyperglycemia or theophylline treatment. Such an approach was intended to provide us with data about long-term disorders in cholinergic neurotransmission, therefore, it was not punctual acetylcholine release, but choline acetyltransferase (ChAT) activity that was analyzed (Figure 2B). As we expected, the cerebellum ChAT activity was almost undetectable, while both treatment strategies resulted in significant downregulation of ChAT activity in the brain septum (Figure 2B).

### 3.2. Isolation and Characterization of the Subcellular Fractions of the SN56 Cells

One of the main concerns about *N*-acetylaspartate production is its localization in the cell. Here, options considered included the mitochondria as the most appreciated localization, then cytosol or/and microsomes as well [40,41,42]. Therefore, in this project, we isolated mitochondrial and cytoplasmic fractions from the SN56 cells to analyze their purity (Table 3). Acute incubation requires SN56 cells to be harvested, centrifuged and then resuspended in depolarizing incubation media. We noted that all of these preliminary steps may enhance the activity of proteolytic enzymes affecting the western blot results. Therefore, studies with mitochondrial fractions were conducted with SN56 cells remaining on the surface of culture dishes for the entire duration of the experiment. Using this approach, the SN56 cells were exposed for 24 h to 0.15 mM Zn^2+^ and analyzed immediately after the experiments reached the final point (Material and Methods, Cell culture).

Although mitochondrial and cytosolic fractions were isolated by established methods, their purity was further verified by western blot and enzymatic assays (Figure 3A,B, Table 3) [19]. Citrate synthase and glutamate dehydrogenase (mitochondrial enzymes) showed the expected activity in the mitochondrial fractions (95% and 90%, respectively), while in cytoplasmic fractions these were almost absent (Table 3). western blot results revealed a similar pattern, where 97% of α-ketoglutarate dehydrogenase protein (a mitochondrial marker) and 9% of glyceraldehyde 3-phosphate dehydrogenase protein (GAPDH, a cytoplasmic marker, loading control) were localized in the mitochondrial fraction (Table 3). In conclusion, the purity of the mitochondrial fraction was greater than 90% and was considered to be sufficient for further studies (Figure 3A,B, Table 3).

### 3.3. Chronic Effect of 0.15-mM Zn^2+^ on the SN56 Cells

Our previous study suggested the strong impact of acetyl-CoA shortages on the NAA level [4]. Other researchers suggest malate–aspartate shuttle disorders as the underlying source of the NAA shortages which have been repeatedly noted in the brains of patients suffering from Alzheimer’s disease [43]. Therefore, the levels of NAA, acetyl-CoA and metabolites involved in the malate–aspartate shuttle were assayed in SN56 cells fractions (Table 4). Both substrates (acetyl-CoA and aspartate) were equally distributed between the mitochondrial and cytoplasmic fractions, while 85% of the total NAA content was assayed in the mitochondria (Table 4). Thus, we considered the mitochondrial content of both substrates to be the main levels controlling NAA production. We noted that acetyl-CoA is highly susceptible to Zn^2+^ toxic influence in both fractions, which results from the Zn^2+^-dependent inhibition of pyruvate dehydrogenase activity, as we have shown previously (Table 4) [4,10,11,12,44]. The other analyzed pathway (the malate–aspartate shuttle) was more resistant to Zn^2+^ - related toxicity. For instance, malate dehydrogenase and aspartate aminotransferase remained resistant to the Zn^2+^ treatment (Table 4). Moreover, lactate dehydrogenase activity (an enzyme controlling the lactate shuttle linked with the malate–aspartate shuttle via NAD/NADH turnover) was not affected either (Table 4). Still, even if the enzymatic profile remains unchanged, the oxaloacetate and aspartate levels dropped by about 30% (Table 4). Further studies of the mitochondrial fraction revealed that deep mitochondrial loss of both acetyl-CoA and aspartate levels was concomitant with the reduction in NAA level (Table 4).

### 3.4. Aspartate N-Acetyltransferase Activity in the SN56 Cells

Table 4 shows the quantitative distribution of NAT8L in the subcellular fractions of the SN56 cells. Similar to glutamate dehydrogenase (a mitochondrial marker), 86% of NAT8L activity was assayed in the mitochondrial fraction (Table 4). The acute 0.15 mM Zn^2+^ treatment suppressed the NAT8L activity by about 25%, although such suppression was observed only in the mitochondrial fraction (Table 4). Considering that the cytoplasmic fraction is contaminated by mitochondria by about 10%, we assumed that the unchanged cytoplasmic NAT8L activity is rather related to the mitochondrial contamination of the cytoplasmic fraction than the actual cytoplasmic NAT8L localization (Table 3 and Table 4)

Since there is no data about the influence of the common divalent transition-metal ions on NAT8L activity, we investigated the impact of Zn^2+^, Cu^2+^, Mn^2+^ using lysed cells SN56 cells (homogenates) (Figure 4E,F). Briefly, 100 µg of cell homogenate protein was incubated in a buffer assay according to the NAT8L assay protocol (Appendix A). In order to analyze the direct impact of Zn^2+^, Cu^2+^ and Mn^2+^ on NAT8L activity, the assay buffer was enriched with these ions in concentrations of up to 0.4-mM (Figure 4E,F). Our data revealed that NAT8L activity is resistant to copper ions, while manganese ions suppress its activity by about 50% (Figure 4E,F). The most powerful concentration-dependent inhibitory effect was noticed for zinc ions counting the [IC50] factor as 0.003 mM (Figure 4E,F).

The final conclusions from chronic and direct (in cells homogenate) approaches revealed that NAT8L is preliminary localized in the mitochondrial fraction of SN56 cells. Furthermore, Zn^2+^ inhibits the NAT8L activity via direct inhibition and affects mitochondrial levels of NAA substrates, which together with NAT8L inhibition, leads to a deep reduction of NAA (Table 4 and Figure 4E,F).

### 3.5. The Acute Effect of 0.15-mM Zn^2+^ on SN56 Cells

In our previous project, voltage-dependent calcium channels were proved to be the regulators of Zn^2+^ toxicity, e.g., the nifedipine treatment limits the free radical overproduction [12]. This time, we wondered whether the Ca^2+^ homeostasis by itself affected NAA production. First, the intracellular Ca^2+^ level was measured, showing its enhanced influx in the Zn^2+^-treated SN56 cells and diminished influx in the nifedipine-treated SN56 cells (Figure 5A). After Zn^2+^ treatment, we noted the significant upregulation of the TBARS level together with the reduction of ATP levels as well as the shortages of acetyl-CoA levels in the SN56 cells and their mitochondrial fraction (Figure 5B and Figure 6A,C,E). Acute treatment of the SN56 cells resulted in Zn^2+^-dependent suppression of the pyruvate dehydrogenase complex, as well as aconitase and isocitrate dehydrogenase activities (Figure 7A,C,E). Further studies showed that the acute Zn^2+^ treatment affected the ATP level first, by the suppression of both pyruvate utilization and lactate production (Figure 8A,C) and second, by the inhibition of the turnover of the malate–aspartate shuttle (Figure 8E,G and Figure 9A,B). In details, the cellular aspartate remained unchanged, although its mitochondrial content was significantly reduced, which indicates the limitation of the malate–aspartate shuttle turnover (Figure 8E and Figure 9A).

The 2-APB antagonist has been reported as a factor triggering the reduction of ATP levels, although the exact mechanism of the 2-APB influence has not been established yet [15]. We noted that 2-APB treatment can disturb acetyl-CoA production by the inhibition of pyruvate dehydrogenase complex activity (Figure 6E and Figure 7A). This costs the tricarboxylic acid cycle its starting substrate (acetyl-CoA, Figure 1A), which leads to the significant inhibition of isocitrate dehydrogenase activity (Figure 6E and Figure 7E). Eventually, the insufficient turnover of the tricarboxylic acid cycle results in ATP shortages (Figure 6C). The other antagonists did not affect these parameters (Figure 6E and Figure 7E). Finally, the NAA level remained resistant to the Mec, NF and 2-APB treatments, although it was significantly reduced by the Zn^2+^ acute treatment (Figure 4A).

### 3.6. The Impact of Hyperglycemia and Theophylline Treatment on Wistar rats’ Bran Septum

Two weeks after the injection with streptozotocin the animals presented deep hyperglycemia, ketoacidosis and overactivation of brain hexokinase activity, while the theophylline treatment did not change these markers significantly (Table 1). Here, to compare the SN56 cells with rats’ brains, we performed the same assays. Neither treatment (streptozotocin, theophylline) significantly affected the lactate dehydrogenase, pyruvate dehydrogenase complex or aspartate aminotransferase activities in both brain regions (Figure 7B,D and Table 5). However, isocitrate dehydrogenase activity was affected by the streptozotocin-induced hyperglycemia in both brain regions (Figure 7F). In the case of metabolic profiles, only the TBARS levels were significantly upregulated in both brain regions after streptozotocin injection (Figure 6B,D,F and Figure 8B,D,F,H). The NAA level in the brain septum remained significantly lower than in the cerebellum and was negatively correlated with choline acetyltransferase activity (Figure 2C and Figure 4B). These findings confirmed our hypothesis that cholinergic neurons prioritized choline neurotransmission over NAA production [4]. Streptozotocin and theophylline challenges significantly reduced NAA level in the cerebellum but did not modify it in the septum brain region (Figure 4B). Our previous study showed that the mature SN56 cells compared to immature cells have significantly higher choline acetyltransferase and NAT8L activities [4]. Therefore, in this study, we analyzed NAT8L activity and *NAT8L* mRNA levels in both brain regions (Figure 4C,D). We learned that even if the brain septum contained less NAA than the cerebellum, NAT8L activity and *NAT8L* mRNA level were similar in both regions (Figure 4C,D). Moreover, the hyperglycemia-dependent upregulation of oxidative stress did not change the NAA level, NAT8L activity or *NAT8L* mRNA level (Figure 4B,D). On the other hand, the theophylline treatment downregulated NAA production (Figure 4B,D).

## 4. Discussion

*N*-acetylaspartate (NAA) deficiency is a common finding associated with brain energy disorders [45,46,47]. These disorders may induce a deficit in sensorimotor gating exhibited by patients suffering from neurodegenerative diseases, such as Alzheimer’s or Parkinson’s disease [45,46,47]. However, little is known about NAA metabolism and its functions in the human central nervous system [4]. In this project we were focused on NAA synthesis and its subcellular distribution in the cholinergic neurons.

Our studies in the SN56 cells homogenates showed that divalent transition-metal ions may exhibit a different inhibition impact against NAT8L activity (Figure 4E,F). For instance, Cu^2+^ is not an NAA synthesis inhibitor, while 0.1 mM Mn^2+^ inhibits this enzyme by almost 50% (Figure 4E,F). Chronic Mn^2+^ neurotoxic effects were studied in the murine thalamus and hypothalamus (25 mg/kg, 3 injection courses within 21 days) [48]. Such treatment affected NAA and glucose levels, which besides the energy-dependent NAA deficiency suggested by the authors could also be explained by the inhibition impact of Mn^2+^ against NAT8L (Figure 4E,F) [48]. Data showing patients suffering from Wilson’s disease (a neurological subpopulation with a high brain Cu^2+^ accumulation) confirmed our observations that NAA production is rather resistant to Cu^2+^-dependent suppression [49]. In our present study, Zn^2+^ was noted as the most efficient dose-dependent inhibitor with a maximal efficiency of over 80% (Figure 4E,F), which fully corresponds with our previous findings showing the toxic influence of Zn^2+^ on NAA level and NAT8L activity [4]. Considering our previous studies and the results from the present in vitro experiments, it could be concluded that Zn^2+^-dependent pathology is related to energy depletion (Figure 5B, Figure 6C,E and Figure 8A,C and Table 4) [4,10,11,12] and inhibition of the cholinergic neurotransmission resulting in the progression of neurodegeneration (Figure 2A) [50,51].

Neurons are initially considered as an exclusive cell type where NAA synthesis takes place, although the subcellular localization of this reaction is still being discussed [52]. Since NAA production is strongly related with the tricarboxylic acid cycle and the malate–aspartate shuttle, we isolated mitochondrial fractions in order to check if they contain NAA (Figure 1A, Table 3; Table 4). About 90% of NAA cellular content and over 85% of NAT8L activity were noted in mitochondria, which we found to be clear evidence that these organelles are responsible for NAA production in SN56 cells (Table 4). Our previous project revealed that the reduction of the NAA level is strongly associated with acetyl-CoA shortages as well as with the inhibition of NAT8L activity and NAT8L protein content [4]. This is consistent with our findings in the present project (Table 4), although in this study we also considered disturbances in the malate–aspartate shuttle as affecting NAA synthesis, as other researchers have suggested [52,53,54]. Two crucial MAS enzymes, malate dehydrogenase (MDH) and aspartate aminotransferase, seem to be resistant to 0.15 mM Zn^2+^-dependent inhibition, which is consistent with the findings reported by other researchers as well [55,56]. However, the mitochondrial level of aspartate was deeply affected by Zn^2+^, both in chronic and acute treatment strategies (Figure 9A and Table 4).

It has been shown that deficiency in the mitochondrial Ca^2+^-regulated aspartate-glutamate carrier isoform 1 (AGC1) decreased the aspartate neuronal level leading to NAA deficiency [43,53]. Little is known about the impact of Zn^2+^ on the malate–aspartate shuttle, but our present project indicates that long-term and acute treatments with 0.15 mM Zn^2+^ affected either cellular or mitochondrial levels of aspartate and oxaloacetate (the main substrates of the malate–aspartate shuttle) in SN56 cells (Figure 8E,G and Figure 9A,B, Table 4). Moreover, our studies showed that the overnight or acute 0.15-mM Zn^2+^ treatment caused significant depletion of NAA level in SN56 cells, especially in the mitochondrial fraction (Figure 4A and Table 3) [4].

In our previous studies, we showed that Zn^2+^ can suppress both NAA and acetyl-CoA levels, although the exact mechanism has not been elucidated [4]. Other studies have reported that nifedipine (NF) can exhibit protective effects against Zn^2+^ toxicity mostly by the prevention of free radical overproduction [12]. Here, using an acute approach, we exposed SN56 cells to nifedipine or Zn^2+^ or nifedipine followed by Zn^2+^. We noted that nifedipine significantly reduced TBARS level, while Zn^2+^ presented the opposite effect (Figure 6A and 1 °C). To elucidate if Zn^2+^-dependent free radical production may affect NAA level, we did studies with nifedipine and Zn^2+^ co-treatment (Figure 10A–D). Indeed, nifedipine prevents Zn^2+^-dependent suppression of NAA (Figure 10A), although further studies showed that such suppression was more likely evoked by the reduction of Zn^2+^ uptake than the antioxidant effect (Figure 10A–D). Moreover, we investigated the impact of 2-APB (an antagonist of the IP3 receptor, intracellular Ca^2+^ release from ER to the mitochondria) on acetyl-CoA and NAA levels (Figure 4A and Figure 6E). One of the commonly observed side effects of 2-APB is the significant depletion of ATP level, also noted in our study (Figure 6C) [15]. Our further studies revealed that such a shortage is associated with a drop in acetyl-CoA availability (Figure 6E). These poor-in-acetyl-CoA conditions did not affect the NAA level, which indicates that the reduction of acetyl-CoA by about 25% did not significantly affect the NAA level (Figure 4A and Figure 2E).

Our previous studies revealed that in cholinergic neurons, *N*-acetylaspartate production is closely related with cholinergic neurotransmission [4]. The maturation of SN56 cells enhanced the acetylcholine level as well as reducing the NAA level [4]. However, it was difficult to establish if the NAA reduction was caused by the higher utilization of acetyl-CoA (to produce acetylcholine) or perhaps such a reduction is related only with the maturation processes. The SN56 cells were differentiated by a mixture of *trans*-retinoic acid with dibutyryl cyclic AMP [4,10,11,12]. It is known that the cellular accumulation of cAMP activates protein kinase A activity, the enzyme controlling CREB-dependent gene expression [57]. It has been reported that the stimulation of protein kinase A may affect the NAA level in the SH-SY5Y human neuroblastoma cell line [57]. Therefore, in this particular study, we used in vivo models to develop the cholinergic neurotransmission disorders in different ways. Firstly, to imitate Zn^2+^-dependent toxicity, we induced chronic hyperglycemia via streptozotocin injection. Here, the animals presented significantly lower choline acetyltransferase activity with significantly higher levels of oxidative stress markers (Figure 6B) [58]. The streptozotocin injection triggered free radical overproduction, although the ongoing oxidative stress did not affect the NAA level in either brain region (Figure 4B). Still, in this model, the hyperglycemia-induced upregulation of oxidative stress had a chronic course; therefore, it is difficult to incontrovertibly state that NAT8L and NAA are resistant to the influence of free radicals. To elucidate this further study with isolated NAT8L enzyme and hydrogen peroxide should be performed. To establish the impact of both disorders in cholinergic neurotransmission and the impact of cyclic AMP accumulation, we challenged the rats for 2 weeks with theophylline. Theophylline is widely known as a compound intensifying cholinergic crisis and as a maturation factor triggering cellular cyclic AMP accumulation [37,59]. The theophylline treatment affected choline acetyltransferase activity, just as streptozotocin injection did (Figure 2B). The inhibition of cholinergic marker activity was more noticeable in the brain septum, although cerebral activity was also affected by both treatment strategies (Figure 2B). However, only theophylline treatment significantly reduced the NAA level as well as NAT8L enzymatic activity and *NAT8L* gene expression (Figure 4B–D). Further studies showed that the higher choline acetyltransferase activity in the brain septum went along with lower NAA production, while in the cerebellum we noted the opposite trend (Figure 2C). Still, NAT8L activity and *NAT8L* mRNA level, acetyl-CoA and aspartate levels were similar in both regions. Therefore, we conclude that the lower NAA level in the brain septum was caused by the prioritizing of acetylcholine production over NAA synthesis (Figure 2B,C). We also noted that the accumulation of cyclic AMP can modify the NAA level not on the level of acetyl-CoA availability, but by the downregulation of NAT8L availability (Figure 4B,D).

## 5. Conclusions

This report provides the first direct evidence for Zn^2+^-dependent suppression of NAA level leading to mitochondrial acetyl-CoA and aspartate shortages as well as by direct concentration-dependent inhibition of NAT8L activity, but was not associated with the upregulation of oxidative stress markers. Since we analyzed the impact of the Zn^2+^ concentration physiologically observed in the synaptic cleft during neurotransmission, we consider that dementia-like diseases concomitant with prolongated neuronal depolarization may indeed develop Zn^2+^-dependent neurotoxicity discussed in this study. Moreover, our results allow us to hypothesize that cholinergic neurons from the brain septum prioritized acetylcholine production over *N*-acetylaspartate production.

## Figures and Tables

**Figure 1 antioxidants-09-00522-f001:**
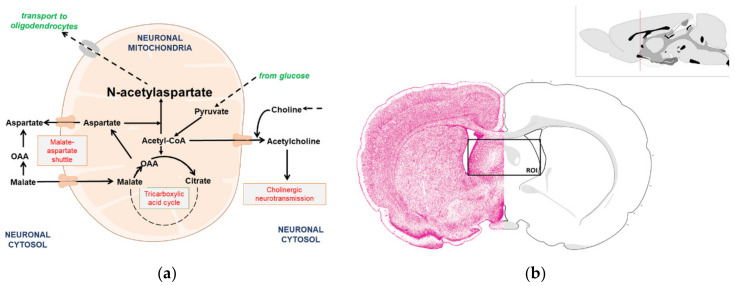
Graphical presentation of the background of this study: (**a**) *N*-acetylaspartate production involves acetyl-CoA and aspartate metabolites; (**b**) representative rat brain section showing the brain septum as a region of interest (ROI) analyzed in this study. Upper image: the cartoon of rat brain sagittal section with red bar indicates the location of the cut used to isolate brain septum tissue. Lower image: resulting cross-section of brain tissue with the brain septum exposed (marked with a red square). Illustrations edited as per author’s permission statement including the Creative Commons Attribution—Noncommercial 4.0 International License regulations [16].

**Figure 2 antioxidants-09-00522-f002:**
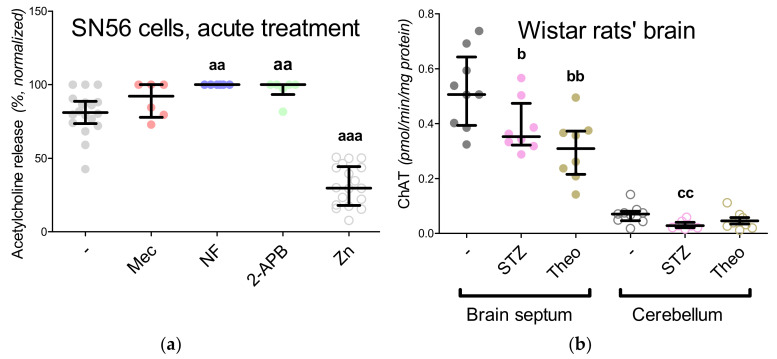
Impact of different experimental conditions on the expression of cholinergic phenotype in (**a**) SN56 cell line (acute experiments, the release of acetylcholine from SN56 cells, data normalized according to the highest value in each set of experiment, average control acetylcholine release: 50.8 pmol/min/mg protein) and (**b**) Wistar rat brain tissue (2 weeks challenge, choline acetyltransferase activity in male adult Wistar rat brain tissue); (**c**) negative correlation between *N*-acetylaspartate level and choline acetyltransferase activity (data calculated from Figure 2B and *N*-acetylaspartate level). Data are median (25th–75th percentile) from 6 to 18 experiments. For in vitro studies, significantly different from SN56 control (^aa^—*p* < 0.01, ^aaa^—*p* < 0.001) or Sham control brain septum (^b^
*p* < 0.05, ^bb^
*p* < 0.01) or Sham control cerebellum (^cc^—*p* < 0.01). Abbreviations: 2-APB—0.050 mM 2-aminoethoxydiphenyl borate; ChAT—choline acetyltransferase; Mec—2 µM mecamylamine; NF—0.01 mM nifedipine; STZ—streptozotocin-induced hyperglycemia; Theo—theophylline; Zn—zinc ions.

**Figure 3 antioxidants-09-00522-f003:**
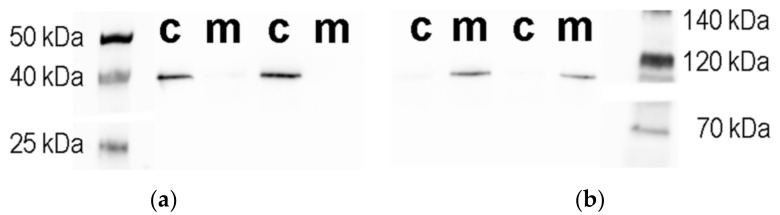
Representative western blot images for (**a**) glyceraldehyde 3-phosphate dehydrogenase or (**b**) α-ketoglutarate dehydrogenase. Abbreviations: c—cytosol fraction; m—mitochondrial fraction.

**Figure 4 antioxidants-09-00522-f004:**
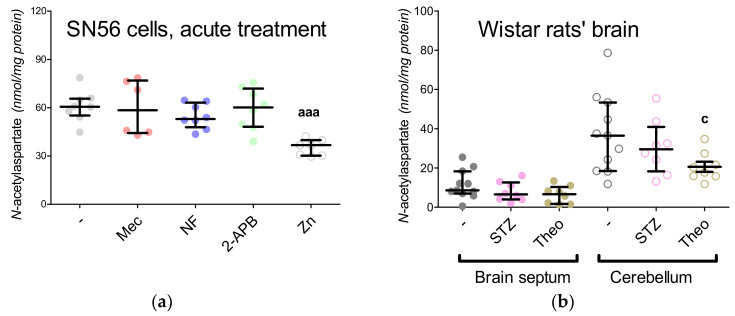
Characteristic features of aspartate N-acetyltransferase (NAT8L) measured in SN56 cells (**a**,**e**,**f**) and Wistar rat brain tissues (**b**–**d**), (**a**,**b**) *N*-acetylaspartate level; (**c**) NAT8L activity; (**d**) NAT8L mRNA level; (**e**) direct effects of divalent transition-metal ions on NAT8L activity assayed in homogenized SN56 cells; (**f**) Dixon’s plot from calculated from data showed at Figure 4e. Data are median (25th–75th percentile) from 6–11 observations. Significantly different from SN56 control (^a^—*p* < 0.05, ^aaa^—*p* < 0.001) or Sham control brain septum (^b^—*p* < 0.05, ^bb^—*p* < 0.01, ^bbb^—*p* < 0.001) or Sham control cerebellum (^c^—*p* < 0.05, ^cc^—*p* < 0.01). Abbreviations: 2-APB—0.050 mM 2-aminoethoxydiphenyl borate, Mec—2 µM mecamylamine, NF—0.01 mM nifedipine STZ: streptozotocin; Theo: theophylline; Zn—zinc ions.

**Figure 5 antioxidants-09-00522-f005:**
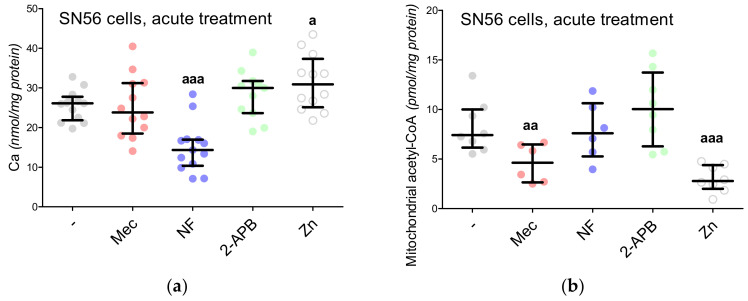
Impact of acute treatment by 0.15-mM Zn^2+^ and the antagonists of calcium-related proteins on SN56 cells: (**a**) intracellular calcium level; (**b**) mitochondrial acetyl-CoA level. Data are median (25th–75th percentile) from 6 to 13 experiments. Significantly different from SN56 control (^a^—*p* < 0.05, ^aa^—*p* < 0.01, ^aaa^—*p* < 0.001). Abbreviations: 2-APB—0.050 mM 2-aminoethoxydiphenyl borate; Ca—calcium ions; Mec—2 µM mecamylamine; NF—0.01 mM nifedipine; Zn–zinc ions.

**Figure 6 antioxidants-09-00522-f006:**
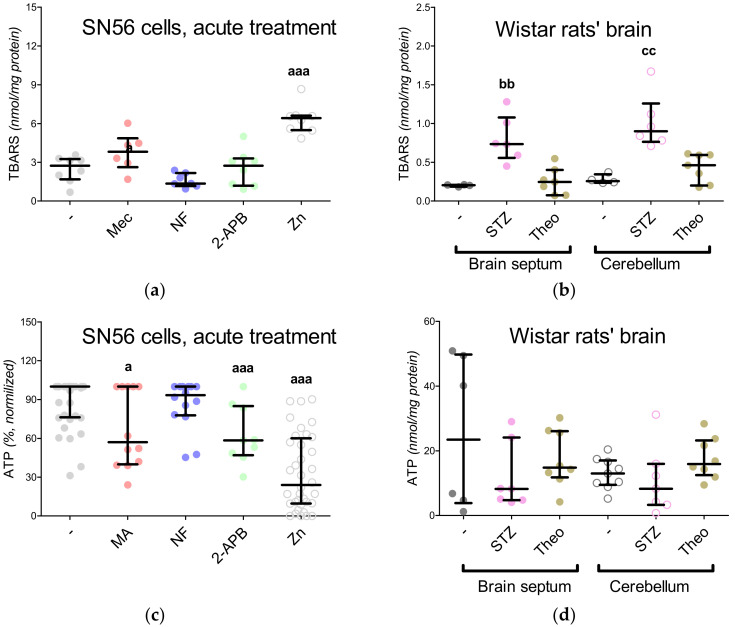
Metabolic profiles of SN56 cells (**a**,**c**,**e**) and Wistar rats’ brain tissues (**b**,**d**,**f**); (**a**,**b**) TBARS level; (**c**,**d**) ATP level (data normalized according to the highest value in each set of experiment, average control ATP level: 6.66 nmol/mg protein); (**e**,**f**) acetyl-CoA level. Data are median (25th–75th percentile) from 4–14 observations. Significantly different from SN56 control (^a^—*p* < 0.05, ^aaa^—*p* < 0.001) or Sham control brain septum (^bb^—*p* < 0.01) or Sham control cerebellum (^cc^—*p* < 0.01). Abbreviations: 2-APB—0.050-mM 2-aminoethoxydiphenyl borate; Mec—2 µM mecamylamine; NF—0.01 mM nifedipine; STZ: streptozotocin; Theo: theophylline; Zn–zinc ions.

**Figure 7 antioxidants-09-00522-f007:**
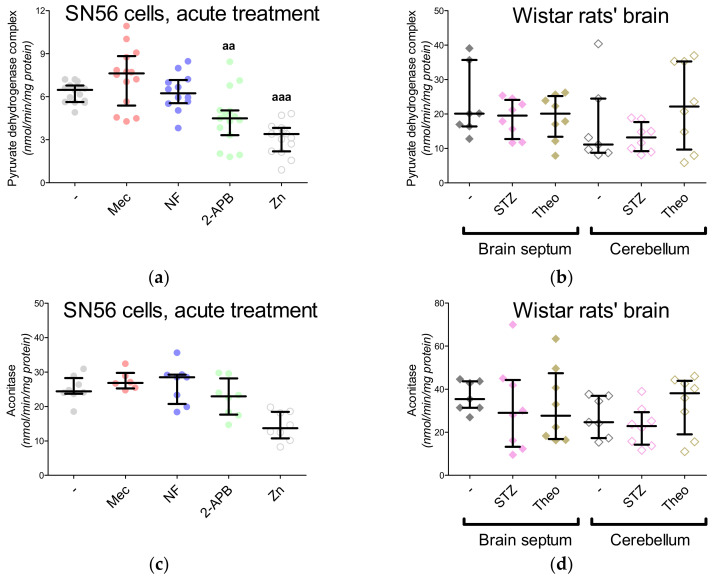
Enzymatic profile of SN56 cells (**a**,**c**,**e**) and Wistar rats’ brain tissues (**b**,**d**,**f**); (**a**,**b**) pyruvate dehydrogenase complex activity; (**c**,**d**) aconitase activity; (**e**,**f**) isocitrate dehydrogenase activity. Data are median (25th–75th percentile) from 6–15 observations. Significantly different from SN56 control (^aa^—*p* <0.01, ^aaa^—*p* < 0.001) or Sham control brain septum (^b^—*p* < 0.05) or Sham control cerebellum (^c^—*p* < 0.05). Abbreviations: 2-APB—0.050 mM 2-aminoethoxydiphenyl borate; Mec—2 µM mecamylamine; NF—0.01 mM nifedipine; STZ: streptozotocin; Theo: theophylline; Zn—zinc ions.

**Figure 8 antioxidants-09-00522-f008:**
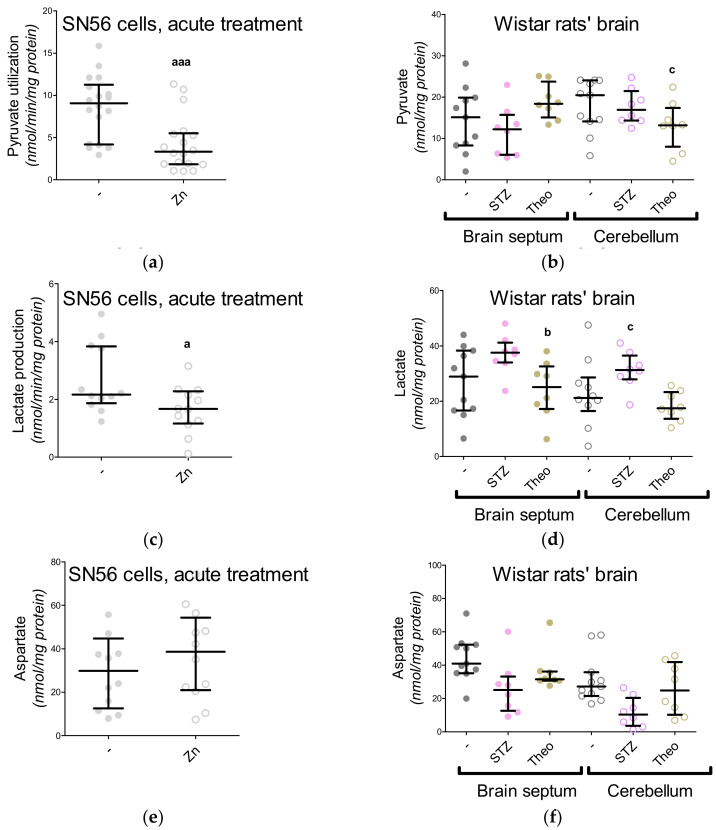
The levels of metabolites linked with malate–aspartate and lactate shuttles measured in SN56 cells (**a**,**c**,**e**,**g**) and Wistar rats’ brain tissues (**b**,**d**,**f**,**h**); (**a**,**b**) pyruvate consumption; (**c**,**d**) lactate production; (**e**,**f**) aspartate level; (**g**,**h**) oxaloacetate level. Data are median (25th–75th percentile) from 8–18 observations. Significantly different from SN56 control (^a^—*p* < 0.05, ^aaa^—*p* < 0.001) or Sham control brain septum (^b^—*p* < 0.05) or Sham control cerebellum (^c^—*p* < 0.05). Abbreviations: STZ: streptozotocin; Theo: theophylline; Zn—zinc ions.

**Figure 9 antioxidants-09-00522-f009:**
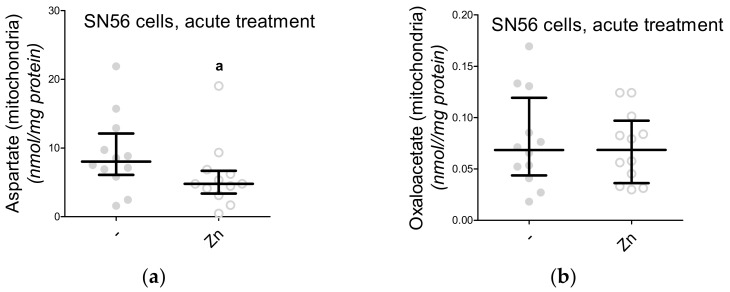
The impact of acute 0.15-mM Zn^2+^ treatment on mitochondrial fractions isolated from SN56 cells: (**a**) mitochondrial aspartate level; (**b**) mitochondrial oxaloacetate level. Data are median (25th–75th percentile) from 12 experiments. Significantly different from SN56 control (^a^—*p* <0.05). Abbreviations: Zn—zinc ions.

**Figure 10 antioxidants-09-00522-f010:**
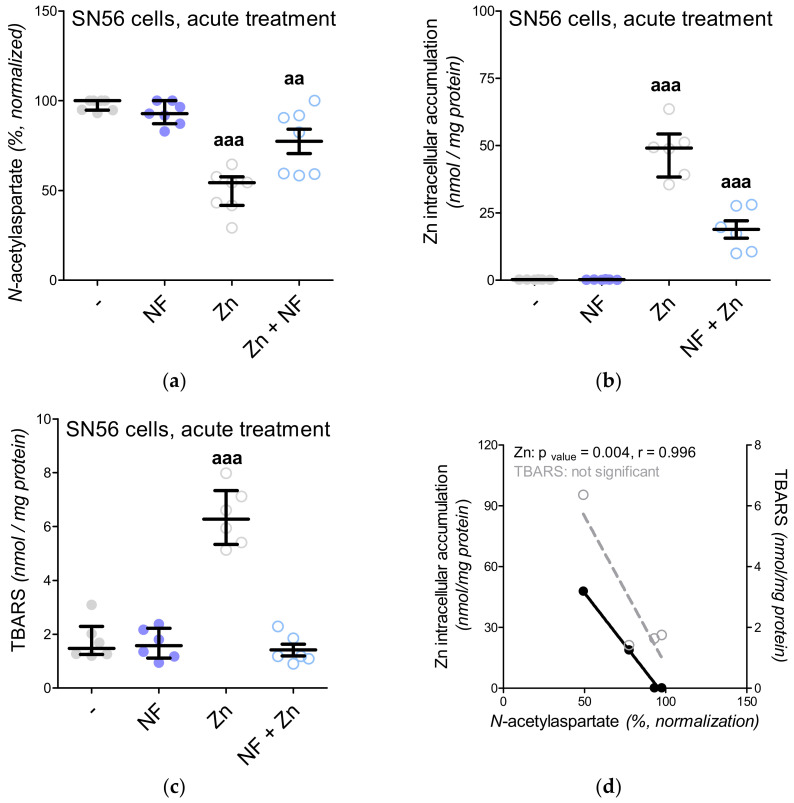
The impact of acute Zn^2+^ and nifedipine treatment on SN56 cell line: (**a**) *N*-acetylaspartate level (data normalized according to the highest value in each set of experiment, average control *N*-acetylaspartate level: 59.1 nmol/mg protein); (**b**) Zn intracellular accumulation; (**c**) TBARS level; (**d**) correlation between *N*-acetylaspartate level and intracellular Zn accumulation (data calculated from Appendix A). Data are median (25th–75th percentile) from 6 to 7 experiments. Significantly different from SN56 control (^aa^
*p* < 0.01, ^aaa^
*p* < 0.001) Abbreviations: NF—0.01 mM nifedipine, Zn–zinc ions.

**Table 1 antioxidants-09-00522-t001:** List of primers and TaqMan probes used in this project.

Parameters	Sham Control	STZ	Theophylline
**Body weight** (g)	290 (275–298)	208 (194–222) ***	310 (296–319) **
**Blood glucose** (mg/dL)	131 (120–133)	518 (500–535) ***	127 (108–141)
**Urine acetoacetate** (µmol/24 h)	0.3 (0.2–0.4)	1.8 (1.0–3.1) ***	0.2 (0.1–0.6)
**Brain hexokinase** (µmol/min/mg protein)	27.6 (27.5–38.4)	62.1 (35.3–92.7) **	28.9 (23.3–96.4)
**Brain β-hydroxybutyrate** (nmol/mg protein)	3.8 (1.6–13.8)	17.1 (16.1–20.7) ***	5.6 (1.5–10.4)

Data are median (25th–75th percentile) from 8–11 animals per group. Significantly different from Sham control (**—*p* < 0.01, ***—*p* < 0.001). Abbreviations: STZ—streptozotocin.

**Table 2 antioxidants-09-00522-t002:** List of primers and TaqMan probes used in this project.

Gene Transcript	Primers	TaqMan Probe	Transcript of Reference Gene
*Nat8l* NM_001191681.1	(F) tggctgacattgaacagtactaca (R) cacaacattgccgtccag	Universal ProbeLibrary Probe #83 (Roche, Cat #04689062001)	Universal ProbeLibrary Rat Actb Gene Assay (Roche, Cat #05046203001)

**Table 3 antioxidants-09-00522-t003:** Characterization of subcellular fractions isolated from SN56 cells (chronic experiments).

Parameters	Whole Cells	Cytoplasmic Fraction	Mitochondrial Fraction	Mitochondria % of Total Value
**Enzymatic Markers**				
**Citrate synthase** nmol/min/mg protein	80.0 (66.3–91.7)	4.7 (1.5–10.2)	77.1 (70.0–81.9)	96
**Glutamate dehydrogenase** nmol/min/mg protein	68.0 (60.1–76.5)	6.3 (4.5–8.7)	62.1 (55.3–66.6)	91
**Western Blot Assay ***				
**Glyceraldehyde 3-phosphate dehydrogenase** peak height value, Au	8.1 (6.6–10.0)	7.3 (5.8–9.4)	0.8 (0.8–0.8)	9
**α-ketoglutarate dehydrogenase** peak height value, Au	8.5 (8.2–9.5)	0.3 (0.2–0.3)	8.3 (7.9–9.3)	97

* images of western blot membranes are attached as Figure 3. Data are median (25th–75th percentile) from 4 to 10 experiments.

**Table 4 antioxidants-09-00522-t004:** Distribution of metabolite levels and enzyme activities in different subcellular fractions isolated from SN56 cells (chronic experiments).

Parameter	Added	Whole Cells	Cytoplasmic Fraction	Mitochondrial Fraction	Mitochondria % of Total Value
**Metabolic Parameters**					
**Acetyl-CoA** pmol/mg protein	Control	27.0 (24.3–31.6)	16.0 (12.2–18.5)	12.3 (11.0–13.5)	45
	0.15-mM Zn^2+^	**12.2** **(10.8–17.8) ^aaa^**	**5.3** **(4.7–10.1) ^aaa^**	**7.2** **(6.1–8.1) ^aaa^**	52
**Oxaloacetate** nmol/mg protein	Control	12.4 (7.2–15.8)	8.3 (3.1–12.6)	4.5 (3.6–5.4)	30
	0.15-mM Zn^2+^	9.1 (5.2–10.1) ^a^	5.2 (2.1–7.3) ^a^	3.4 (2.5–4.0) ^a^	26
**Aspartate** nmol/mg protein	Control	60.9 (46.7–73.1)	37.1 (14.2–44.5)	28.8 (21.5–31.6)	47
	0.15-mM Zn^2+^	43.1 (36.7–50.8) ^a^	32.9 (28.3–42.1)	8.3 (6.5–11.2) ^aaa^	19
**Malate** nmol/mg protein	Control	16.2 (11.5–20.1)	8.9 (5.9–14.6)	6.4 (5.0–7.1)	40
	0.15-mM Zn^2+^	14.0 (11.2–18.1)	7.8 (4.2–11.4)	6.5 (5.0–8.0)	46
***N*-acetylaspartate** nmol/mg protein	Control	62.8 (52.5–73.7)	6.7 (5.2–18.1)	53.2 (47.7–58.4)	85
	0.15-mM Zn^2+^	30.2 (28.2–35.0) ^aaa^	2.2 (1.5–4.4) ^aa^	27.0 (26.3–30.7) ^aaa^	89
**Enzymatic Parameters**					
**NAT8L** pmol/min/ mg protein	Control	82.4 (59.7–88.8)	6.5 (5.2–13.3)	69.6 (47.3–76.7)	84
	0.15-mM Zn^2+^	55.5 (46.1–73.0) ^a^	10.5 (7.2–16.9)	38.6 (31.4–76.7) ^aa^	70
**Malate dehydrogenase** µmol/mig/mg protein	Control	0.7 (0.6–0.8)			
	0.15 mMZn^2+^	0.6 (0.6–0.7)			
**Aspartate aminotransferase** nmol/min/mg protein	Control	56.0 (52.9–71.3)			
	0.15-mM Zn^2+^	53.6 (50.2–59.0)			
**Lactate dehydrogenase** µmol/min/mg protein		1.7 (1.2–2.1)			
		2.0 (1.6–2.3)			

Data are median (25th–75th percentile) from 8–13 observations. Significantly different from related control (^a^—*p* < 0.05, ^aa^—*p* <0.01, ^aaa^—*p* < 0.001). Abbreviations: NAT8L—aspartate N-acetyltransferase.

**Table 5 antioxidants-09-00522-t005:** Enzymatic activities measured in Wistar rats’ brain.

Parameter	Added	Wistar rats’ Brain	
		Brain Septum	Cerebellum
**Enzymatic Parameters**			
**Aspartate aminotransferase** nmol/min/mg protein	Control STZ Theophylline	1.0 (0.8–2.0) 0.9 (0.4–1.8) 1.4 (0.9–1.7)	1.0 (0.9–1.4) 0.7 (0.4–1.3) 1.6 (0.8–2.2)
**Lactate dehydrogenase** µmol/min/mg protein	Control STZ Theophylline	0.4 (0.3–0.4) 0.4 (0.3–0.5) 0.3 (0.2–0.5)	0.4 (0.3–0.6) 0.4 (0.2–0.5) 0.3 (0.3–0.6)

Data are median (25th–75th percentile) from 7 to 10 experiments.

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
