# Peer review of "The Impact of Acetyl-CoA and Aspartate Shortages on the N-Acetylaspartate Level in Different Models of Cholinergic Neurons"

_antioxidants, 2020, doi:10.3390/antiox9060522_

Round 1
Reviewer 1 Report
In this work the authors analyze the effect of toxic conditions on the N-acetyl aspartate (NAA) production. They use three different models one in vitro and two in vivo. Their model in vitro is the SN56 cells, a cellular model of cholinergic neurons, that they challenge with Zn2+ or Ca2+ dysregulation. The in vivo models are Wistar rats treated with streptozotocin to induced hyperglycemia or daily theophylline treatment.
They report that Zn2+ toxicity suppresses NAA production via the shortage of mitochondrial acetyl-CoA and aspartate and the inhibition of the neuronal aspartate N-acetyltransferase (NAT8L).
The manuscript is presenting an analysis of the pathways leading to NAA production in a fairly detailed way they use a cell line and two induced pharmacological models. What is not clear to me is the link between the in vitro and in vivo models, since to my understanding, the in vivo models are not examples of neurodegenerative diseases and in the conclusions the authors don’t mention at all the two models. What was the purpose of using it? The authors should clarify this in the manuscript since, in my opinion, it is not clear at all.
I think it would be helpful for the reader to see in a cartoon, for example the exact septal area analyzed in the study and they should describe in more detail how they isolate the septum.
The manuscript needs a revision of the English, some sentences are difficult to understand.
Author Response
The authors wish to extend their appreciation to the Reviewer’s considering this Manuscript as potentially suitable to publish at Antioxidants. The Manuscript was improved in accordance with the Reviewer’s comments. To see the exact changes, please read the Manuscript using the “track changes” option.
Reviewer #1
Reviewer #1: The manuscript is presenting an analysis of the pathways leading to NAA production in a fairly detailed way they use a cell line and two induced pharmacological models. What is not clear to me is the link between the in vitro and in vivo models, since to my understanding, the in vivo models are not examples of neurodegenerative diseases and in the conclusions the authors don’t mention at all the two models. What was the purpose of using it? The authors should clarify this in the manuscript since, in my opinion, it is not clear at all.
Corresponding author’s response: The authors agreed that the connections between in vitro and in vivo models were not discussed properly, therefore we clarified it in lines: 71-91, 531-544 and 548-555.
Reviewer #1: I think it would be helpful for the reader to see in a cartoon, for example the exact septal area analyzed in the study and they should describe in more detail how they isolate the septum.
Corresponding author’s response: The authors agreed with the Reviewer’s comment and improved the Methods (lines: 124-128) as well as by adding additional figure (Figure 1, lines 92-97).
Reviewer #1: The manuscript needs a revision of the English, some sentences are difficult to understand.
Corresponding author’s response: Considering the Reviewer's kind comment, the Manuscript was verified by the language proofreading agency.
Reviewer 2 Report
The authors described in this manuscript a more extended analysis of their previous findings reported in 2017 and 2018 on Zn toxicity on NAA levels, particularly the inclusion of rat brain data. On the whole the analyses are carefully done and added to knowledge in the field. The authors' conclusion that Zn directly inhibits NAA synthesis via suppression of neuronal aspartate N-acetyltransferase activity is also an important point to make.
Major issues:
- Zn could inhibit multiple mitochondrial enzymes, and the authors claim to have excluded a role of oxidative stress in Zn's suppression of NAA, acetyl-CoA and aspartate N-acetyltransferase activity based on streptozotocin-induced hyperglycemia. Is there more direct evidence for this notion, for example a reducing agent like NAC could not alleviate the inhibition?
- The consequence of Zn toxicity in the animals are under-described. The authors examined choline acetyltransferase (ChAT) activity, but are the animals impaired in feeding and behavior that might perturb their brain metabolite levels?
Other issues:
As this is an online journal with no real page limits, this reviewer does not see the need to have supplementary figures. In other words, all the figures should be shown upfront for the easy access by the reader.
Author Response
The authors wish to extend their appreciation to the Reviewer’s considering this Manuscript as potentially suitable to publish at Antioxidants. The Manuscript was improved in accordance with the Reviewer’s comments. To see the exact changes, please read the Manuscript using the “track changes” option.
Reviewer #2
Reviewer #2: Zn could inhibit multiple mitochondrial enzymes, and the authors claim to have excluded a role of oxidative stress in Zn's suppression of NAA, acetyl-CoA and aspartate N-acetyltransferase activity based on streptozotocin-induced hyperglycemia. Is there more direct evidence for this notion, for example a reducing agent like NAC could not alleviate the inhibition?
Corresponding author’s response: The authors agreed with the Reviewer’s comment and clarified this aspect (lines: 542-548).
Reviewer #2: The consequence of Zn toxicity in the animals are under-described. The authors examined choline acetyltransferase (ChAT) activity, but are the animals impaired in feeding and behavior that might perturb their brain metabolite levels?
Corresponding author’s response: The most efficient way to induce Zn2+-dependent toxicity in brain is to microinject the particular brain regions, which was technically impossible in our laboratory. However, we did consider the Zn2+-rich diet as a resolution, although such approach has been reported as a deeply affecting the internal organs (e.g. liver) instead of brain. Therefore, for in vivo studies, we decided to induce the cholinergic neurons degeneration via chronic hyperglycaemia or theophylline treatment, which as Reviewer pointed was controlled by ChAT activity. In this study, we did not perform behavioural assessment, although we noted streptozotocin – induced rat being overactive during the first 7 days after injection. Since such behaviour has been reported by other researchers as a typical hyperglycaemia – related behaviour, no further studies have been performed. However, we did the feeding assessment with metabolic cage. Streptozotocin – induced animals presented polydipsia and polyuria, while the theophylline – treated animals had similar results as the sham controls.
Reviewer #2: Other issues: As this is an online journal with no real page limits, this reviewer does not see the need to have supplementary figures. In other words, all the figures should be shown upfront for the easy access by the reader.
Corresponding author’s response: The Manuscript has been improved according to the Reviewer’s comment. The data from Table 2S has been either incorporated to Table 4 (enzymatic profile of SN56 cells, lines: 337-340) or presented as Table 5 (enzymatic profile of Wistar rats’ brain, Lines: 467-468). Figured were moved from supplementary data to the Manuscript having a new numbers as follows: Figure 1 S is now Figure 3 (Lines: 273-284), Figure 2 S is now Figure 2 (Lines: 307-317), Figure 3S in now Figure 5 (Lines: 404-410), Figure 4S is now Figure 9 (Lines: 442 - 446), Figure 5S is now Figure 10 (Lines: 564 - 572).
Reviewer 3 Report
The authors present an extensive study on the balance of acetylcoA between several signalling and metabolic pathways: acetylcholine, N-acetylaspatate and tricarboxylic acids cycle. They include both cell and in vivo models with different conditions and asses the level of various markers and enzymatic activities.
*The experiments are correctly done but the results could be presented more clearly. There are sentences which meaning are obscure, for instance, in line 315-317 "Here, even if the wole pools..".
The last sentence of this section is also not very clear 'line 326-328): "SN56 cells can handle acetylcoA shortages and keeeeps the NAA level stable." THe study shows that the level of NAA varies quite much...
*Supplementary figures are not correctly referenced:
line 307 and 312: Figure 3S instead of 2S
line 315: figure 4S
*"two-charges divalent transient metal ions" should be changed bi "divalent transition-metal ions" in lines 289 and 400
*calculation of statistical significance between brain setup and cerebellum should be done by taking control of each group as reference, ie control each region for calculation with STZ and theo respectively. Cross comparison between brain septum and cerebellum could be done in another test.
*There are some minor spelling errors. For instance in line 311 "the ATP cellular levels of ATP"
line 373 "compere"
Author Response
REBUTTAL LETTER
The authors wish to extend their appreciation to the Reviewer’s considering this Manuscript as potentially suitable to publish at Antioxidants. The Manuscript was improved in accordance with the Reviewer’s comments. To see the exact changes, please read the Manuscript using the “track changes” option.
Reviewer #3
Reviewer #3: The last sentence of this section is also not very clear 'line 326-328): "SN56 cells can handle acetylcoA shortages and keeeeps the NAA level stable." THe study shows that the level of NAA varies quite much...
Corresponding author’s response: The Authors agreed with Reviewer’s comment, therefore conclusion about the stability of NAA level in SN56 cell line was removed.
Reviewer #3: *Supplementary figures are not correctly referenced:
line 307 and 312: Figure 3S instead of 2S
line 315: figure 4S
Corresponding author’s response: The authors agreed with the Reviewer’s comments and improved the figures’ order. All figures have been moved to the main body of the manuscript as well as got a new number order The data from Table 2S has been either incorporated to Table 4 (enzymatic profile of SN56 cells, lines: 337-340) or presented as Table 5 (enzymatic profile of Wistar rats’ brain, Lines: 467-468). Figured were moved from supplementary data to the Manuscript having a new numbers as follows: Figure 1 S is now Figure 3 (Lines: 273-284), Figure 2 S is now Figure 2 (Lines: 307-317), Figure 3S in now Figure 5 (Lines: 404-410), Figure 4S is now Figure 9 (Lines: 442 - 446), Figure 5S is now Figure 10 (Lines: 564 - 572).
Reviewer #3: *The experiments are correctly done but the results could be presented more clearly. There are sentences which meaning are obscure, for instance, in line 315-317 "Here, even if the wole pools..".
*"two-charges divalent transient metal ions" should be changed bi "divalent transition-metal ions" in lines 289 and 400
*There are some minor spelling errors. For instance in line 311 "the ATP cellular levels of ATP"
line 373 "compere"
Corresponding author’s response: The Manuscript was improved according to the Reviewer comments. Additionally, the Manuscript was verified by the language proofreading agency.
Reviewer #3: *calculation of statistical significance between brain setup and cerebellum should be done by taking control of each group as reference, ie control each region for calculation with STZ and theo respectively. Cross comparison between brain septum and cerebellum could be done in another test.
Corresponding author’s response: The Manuscript was improved according to the Reviewer comments. Calulations were performed on data presented at Figures: 2 (lines: 272-284), 4 (lines: 364-375), 6 (lines: 409-417), 7 and 8 (lines: 420-439).
Round 2
Reviewer 2 Report
The authors have improved their manuscript. There are only minor points that could be further addressed.
- Instead of being dogmatic and state that `Zn2+-triggered oxidative stress is rather not involved...' (abstract), it might be better to have a more reserved statement like 'Zn2+-triggered oxidative stress is unlikely to be significant in...'.
- For the western blot in Fig 3, the molecular size marking shown should point (with lines or arrows) towards a position of the blot (in other words, it should be a true marking in the experiment and not a predicted molecular size of the bands).
- For Fig 1a, the main cast N-acetylaspartate should stand out in a schematic like such. Fig 1b legend needs to be more complete. We see brain coronal and sagittal section diagrams (should be indicated) and the box and red line should be briefly explained.
Author Response
Reviewer #2
Reviewer #2: The authors have improved their manuscript. There are only minor points that could be further addressed.
- Instead of being dogmatic and state that `Zn2+-triggered oxidative stress is rather not involved...' (abstract), it might be better to have a more reserved statement like 'Zn2+-triggered oxidative stress is unlikely to be significant in...'.
Corresponding author’s response: The Abstract was improved according to the Reviewer comments (lines: 31-32).
Reviewer #2:
- For the western blot in Fig 3, the molecular size marking shown should point (with lines or arrows) towards a position of the blot (in other words, it should be a true marking in the experiment and not a predicted molecular size of the bands).
Corresponding author’s response: The Figure 3 was improved according to the Reviewer comments (lines: 309-310).
Reviewer #2:
- For Fig 1a, the main cast N-acetylaspartate should stand out in a schematic like such. Fig 1b legend needs to be more complete. We see brain coronal and sagittal section diagrams (should be indicated) and the box and red line should be briefly explained.
Corresponding author’s response: The Figure 1 was improved according to the Reviewer comments (lines: 92-99).